# Suicidal ideation, attempt, and its associated factors among adult HIV/AIDS patients in Ethiopia: A systematic review and meta-analysis study

Eyob Ketema Bogale[ID][1]*, Amare Zewdie[2], Tadele Derbew Kassie[ID][3], Tadele Fentabil Anagaw[1], Elyas Melaku Mazengia[3], Sintayehu Shiferaw Gelaw[3], Eneyew Talie Fenta[4], Habitu Birhan Eshetu[5], Natnael Kebede[6]

1 Department of Health Promotion and Behavioral Science, School of Public Health, College of Medicine and Health Science, Bahir Dar University, Bahir Dar, Ethiopia, 2 Department of Public Health, Collage of Medicine and Health Science, Wolkite University, Wolkite, Ethiopia, 3 Department of Public Health, College of Medicine and Health Science, Debre Markos University, Debre Markos, Ethiopia, 4 Department of Public Health, College of Medicine and Health Sciences, Injibara University, Injibara, Ethiopia, 5 Department of Health Promotion and Health Behaviour, Institute of Public Health, College of Medicine and Health Sciences, University of Gondar, Gondar, Ethiopia, 6 Department of Health Promotion, School of Public Health, College of Medicine and Health Sciences, Wollo University, Dessie, Ethiopia

* ketema.eyob@gmail.com

**Data Availability Statement:** All relevant data are within the paper.

## Abstract

### Background

WHO statistics show that someone attempts suicide every three seconds and commits suicide every 40 seconds somewhere in the world. There is a scarcity of aggregate evidence in Ethiopia. The aim of this review was to assess the pooled prevalence of suicidal ideation, attempts, and associated factors among adult HIV/AIDS patients in Ethiopia to fill this gap.

### Methods

We extensively searched the bibliographic databases of PubMed, MEDLINE, Scopus, Google Scholar, and the Web of Science to obtain eligible studies. Further screening for a reference list of articles was also done. The Microsoft Excel Spreadsheet was used to extract data, and Stata 17 was used for analysis. To check heterogeneity, the Higgs I2 and Cochran's Q tests were employed. Sensitivity and subgroup analysis were implemented. To detect publication bias, Egger's test and funnel plots were used.

### Results

The pooled prevalence of suicidal ideation and attempts among adult HIV/AIDS patients in Ethiopia was 20.3 with a 95% CI (14, 26.5) and 11.1 with a 95% CI (6.6, 15.5), respectively. Living alone (AOR 4.98; 95% CI: 2.96–8.37), having comorbidity or other opportunistic infection (AOR 4.67; 95% CI: 2.57–8.48), female sex (AOR 2.86; 95% CI: 1.76, 4.62), having WHO clinical stage III of HIV (AOR 3.69; 95% CI: 2.15, 6.32), having WHO clinical stage IV of HIV (AOR 5.43; 95% CI: 2.81, 10.53), having co-morbid depression (AOR 5.25; 95% CI:

**Funding:** The author(s) received no specific funding for this work.

**Competing interests:** The authors have declared that no competing interests exist.

**Abbreviations:** AIDS, Acquired Immune-Defciency Syndrome; AOR, Adjusted Odd Ratio; CI, Confdence Interval; CIDI, Composite International Diagnostic Interview; DF, Degree of Freedom; HIV, Human Immune Virus.

4.05, 6.80), having perceived HIV stigma (AOR 2.53; 95% CI: 1.67, 3.84), and having family history of suicidal attempt (AOR 2.79; 95% CI: 1.38, 5.66) were significantly associated with suicidal ideation. Being female (AOR 4.33; 95% CI: 2.36, 7.96), having opportunistic infections (AOR 2.73; 95% CI: 1.69, 4.41), having WHO clinical stage III of HIV (AOR 3.78; 95% CI: 2.04, 7.03), having co-morbid depression (AOR 3.47; 95% CI: 2.38, 5.05), having poor social support (AOR 3.02; 95% CI: 1.78, 5.13), and having WHO clinical stage IV (AOR 7.39; 95% CI: 3.54, 15.41) were significantly associated with suicidal attempts.

## Conclusion

The pooled magnitude of suicidal ideation and attempt was high, and factors like opportunistic infection, WHO clinical stage III of HIV, WHO clinical stage III of HIV, and co-morbid depression were related to both suicidal ideation and attempt. Clinicians should be geared towards this mental health problem in HIV patients during management.

## Introduction

The definition [1] states that suicide is defined as deliberate self-inflicted death. A suicidal attempt is a deliberate but unsuccessful attempt to kill oneself, and suicidal ideation is having thoughts of doing so [1]. According to WHO (World Health Organization) statistics, someone attempts suicide every three seconds and commits suicide every 40 seconds somewhere in the world. At least six additional people are severely impacted psychologically, socially, and financially by a single suicide, and suicide represents 1% of the global disease burden [2].

By the year 2020, the yearly global rate of suicide will rise to 11.4 per 100,000 people, accounting for nearly 2.4% of the world's illness burden, and there will be one suicide-related death every 20 seconds. In 2013, the cost of suicide in the United States, including suicide attempts, was $58.4 billion [3].

According to a self-report questionnaire used in an institutional-based cross-sectional study in Nanjing, China, patients with HIV/AIDS have a suicide ideation rate of 31.6% [4]. In the USA, a cross-sectional study with 1560 individuals found that the prevalence of suicidal thoughts and attempts was 26% and 13%, respectively [5].

A systematic review and meta-analysis study report revealed that the pooled prevalence of suicidal ideation and attempts among patients with HIV/AIDS in Africa was 21.7% and 11.06%, respectively [6]. According to a study done in Kampala, Uganda, the frequency of suicidal ideation is 17%, and suicidal ideation is more common than suicide attempts [7]. Another cross-sectional study comprising 109 pregnant women in the second half of their pregnancy and a sizable primary health care center in a rural area of KwaZulu-Natal, South Africa, revealed a 27.5% suicidal ideation rate [8].

According to the results of a study on suicidality among HIV patients at a treatment facility in the Nigerian city of Kaduna, the prevalence of suicidality was 16% among the study's 250 total participants [9]. The frequency of suicidal thoughts was found to be 26.9% among adolescents in Tunisia, and the suicide attempt rate was 7.3% [10]. The proportion of suicidal thoughts and attempts was found to be 33.6% and 20.1%, respectively, in hospital-based cross-sectional research among HIV/AIDS patients at Debark District Hospital North-West, Ethiopia [11].

Suicidal thoughts and attempts are high. The size of suicidal ideation was found to be 27.1% [12] in a study conducted at St. Paul's Hospital, Millennium Medical College, and St. Peter's Specialized Hospital. Suicidal attempts were reported to be 16.9% in a related study. According to a study done in the same city with 423 study participants, the prevalence of suicidal ideation and attempts was 22.5% and 13.9%, respectively [13]. Another cross-sectional study on suicidal ideation among HIV/AIDS patients at northeastern Ethiopia's Dessie Referral Hospital found that the prevalence was 9.4%, with 3.3% of those having attempted suicide [14].

The majority of HIV-infected adolescents live in environments with inadequate resources, and as a result, they frequently struggle with mental and behavioral health issues, which have an impact on all facets of HIV prevention and treatment. As a result, such mental health issues cause suicidal thoughts and behaviors. The effects of mental health issues on young people living with HIV have not been sufficiently studied [15].

In Ethiopia, youth and young adults account for a large percentage of all HIV/AIDS cases. Hence, living with chronic diseases like HIV/AIDS may increase the risk of suicide. There is no systematic review or meta-analysis report of suicide among HIV-positive adults in Ethiopia, and the results of those studies have sometimes been contradictory when it comes to the causes of suicidal thoughts and attempts. Therefore, it is necessary to evaluate high-risk populations for suicide ideation early on and to provide PLWHA (people living with HIV/AIDS) with greater psychological health care. The findings of previous studies that differ from one another require systematic evaluation and meta-analysis. As a result, the objective of this systematic review and meta-analysis study is to evaluate the combined prevalence of suicidal ideation and attempt and its associated factors among HIV-positive adults in Ethiopia.

## Methods

### Search strategy

This systematic review and meta-analysis study was undertaken following the preferred Reporting Items for Systematic Reviews and Meta-Analyses (PRISMA) guidelines [16]. This systematic review and meta-analysis study was registered in PROSPERO with reference ID CRD42023439054. This review study took place between June 1, 2023, and June 30, 2023.

The search strategy for this review has been done in two ways. The first was an exploration of electronic databases (PubMed, Scopus, and Google Scholar) for the presence of evidence regarding suicidal ideation and attempts in HIV/AIDS patients. We have used the following headings and keywords for PubMed database searching: prevalence OR magnitude AND "suicidal ideation" OR suicidality OR "suicidal attempt" AND HIV OR AIDS OR ART AND factor OR "risk factor" OR determinant OR predictors AND Ethiopia. Google Scholar and Scopus were conducted in line with database-specific searching guidelines using keywords used in PubMed. The second strategy was a manual search for the reference lists of the incorporated studies. We had put 2013–2023-time restrictions on our search for articles.

### Inclusion and exclusion criteria

**Inclusion criteria.** An article was eligible for inclusion if it meets the following criteria:

- The study had been conducted in adult HIV/AIDS patients,

- The outcome investigated should be suicidal ideation, suicidal attempt and its associated factors

- The study should be conducted in Ethiopia.

- The study conducted from 2013 to 2023,

**Exclusion criteria.** An article was eligible for exclusion if it meets the following criteria:

- Previous studies which assessed non-human subjects.

- Publication of the article in Non-English language.

## Methods for data extraction and quality assessment

The three authors (EKB, ET, and AZ) independently extracted the essential data from the studies included in the final analysis using a standardized data extraction template. A standardized Microsoft Excel spreadsheet was used to enter the data that had been extracted from the included research. To maintain consistency, cross-checking was done. During the extraction process, discrepancies between the extracted data were resolved through logical discussion among the three authors, and the final consensus was approved with the participation of authors NK and HBE.

The table includes the author's name, the research population, the sample size, the publication year, the study area, the study design, and the evaluation tool for suicidal thoughts and attempts and its associated factors in HIV patients.

Utilizing a PRISMA-compliant assessment form, the data from the included studies was collected [16]. The Newcastle-Ottawa quality assessment scale [17] was used to rate the caliber of the research that was considered in the final analysis. This scale allows for possible scores ranging from 0 to 10. On this scale, a score of 8 or more indicated good quality, a score of 3 to 7 indicated moderate quality, and a score of less than this indicated low quality. Two independent authors (EKB and TF) assessed the included studies' quality using the Newcastle-Ottawa quality assessment scale [17]. The third author (NK) discussed and resolved any disagreements. Out of the seven studies that were considered, two had moderate quality and five had good quality.

## Data synthesis and analysis

In this meta-analysis, we used a random-effects model to calculate the pooled prevalence of suicidal ideation and attempt and its associated factors with their 95% CIs. STATA 17 was used for this meta-analysis method. When the I2 statistical value is zero, there is no heterogeneity; I2 values of 25, 50, and 75%, respectively, imply small, moderate, and large heterogeneity [18].

Potential heterogeneity had an impact on our study; therefore, we did a sensitivity analysis to identify its most likely source. In order to assess suicidal ideation and attempts, a subgroup analysis based on sample size, publication year, and study setting had to be completed. Two tests—the ocular funnel plot test and Egger's regression test—were used to determine whether there was publication bias. P-values < 0.05 were used to define statistical significance for analyses.

## Results

### Identifcation of studies

A total of 558 papers were found during our search for literature utilizing the previously mentioned search techniques. A second search for the reference list of publications that were supported by references also turned up 13 articles, for a total of 571 articles. Before the screening, a total of 106 articles were excluded due to duplicate records (n = 94), records marked as ineligible by automation tools (n = 13), and records removed for other reasons (n = 9).

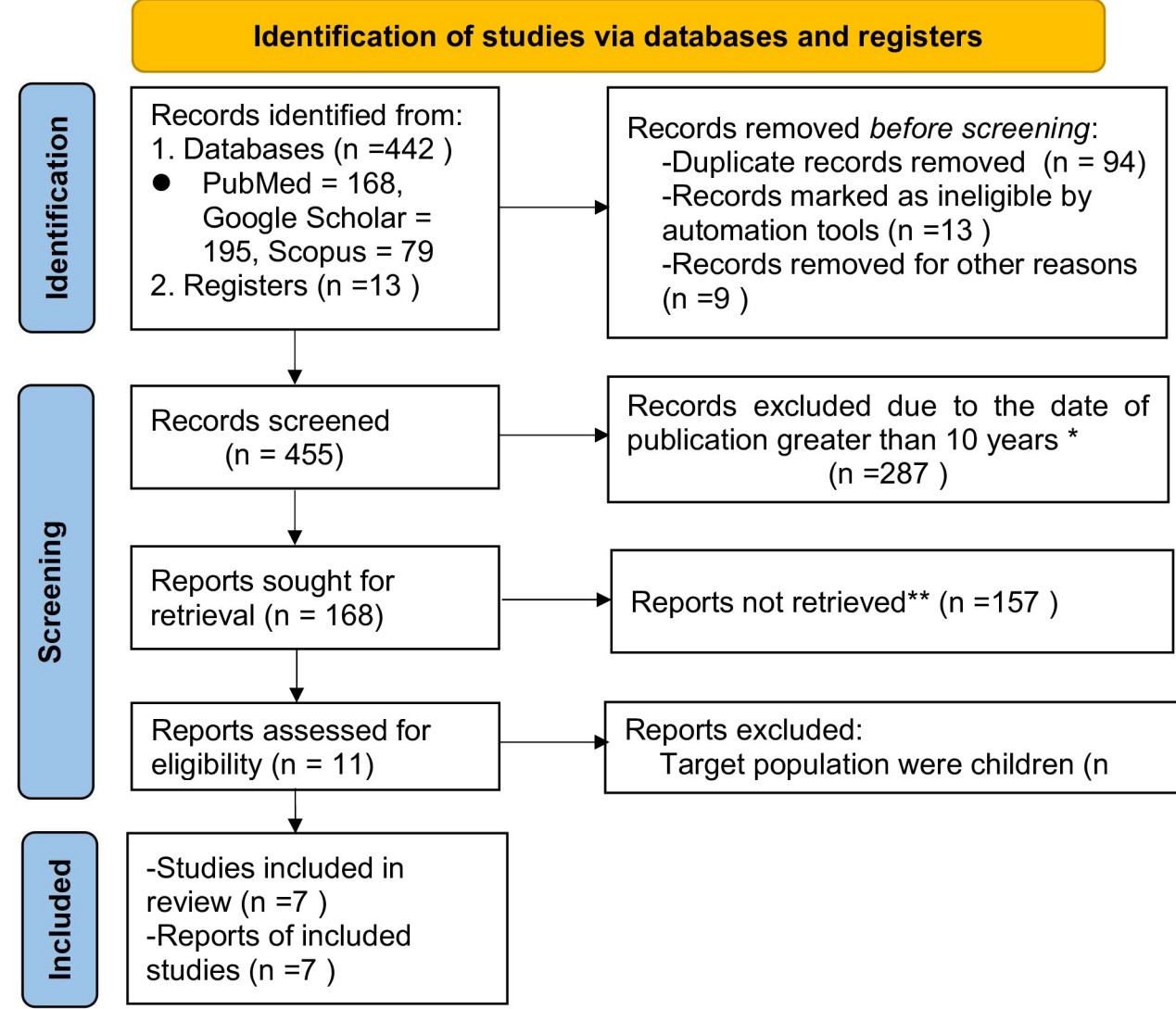

*Date of publication greater than 10 years, **Target population were children

**Fig 1. Flow chart for the review process.** * Data of publication greater than 10 years, ** Target population were children.

During and after screening, a total of 157 of these articles were excluded since their publication dates were more than ten years in the past. A total of 287 papers were eliminated by simply reading their titles (86 were eliminated because the target group was not HIV/AIDS patients, and 201 were eliminated because the title hinted that the end variable was not suicidal ideation or attempt). The remaining 11 studies were thoroughly examined to see whether they should be included in the meta-analysis; however, only 7 articles [11–14, 19–21] were included since the remaining 4 articles were also excluded because the meta-analysis's target population age was children (Fig 1).

## Characteristics of included studies

An overall of 7 studies that examined suicidal ideation or suicidal attempts in 2593 HIV/AIDS adult patients in Ethiopia were included in this systematic review and meta-analysis study

**Table 1. Characteristics of studies on sucidial ideation and sucidial attempt among adult HIV/ AIDS patients in Ethiopia.**

| Sr. no | Author/year | Study area | Study design | Sample size | Tool used | Out come variable | % Sucidial ideation | % Sucidial Attempt |
|---|---|---|---|---|---|---|---|---|
| 1 | Tesera Kendie M, et al, 2023 | Addis Ababa | CS | 237 | **CIDI** | Sucidial ideation and attempt | 22.8 | 13.5 |
| 2 | Wonde M. et al, 2019 | Addis Ababa | CS | 423 | **CIDI** | Sucidial ideation and attempt | 27.1 | 16.9 |
| 3 | Bitew H. et al, 2016 | Debark | CS | 393 | **CIDI** | Sucidial ideation and attempt | 33.6 | 20.1 |
| 4 | Tamirat KS. et al, 2021 | Dessie | CS | 395 | **CIDI** | Sucidial ideation and attempt | 9.4 | 3.3 |
| 5 | Gebremariam EH. et al, 2017 | Addis Ababa | CS | 423 | **CIDI** | Sucidial ideation and attempt | 22.5 | 13.9 |
| 6 | Gebreegziabhier Kindaya G, 2020 | Harar | CS | 423 | **CIDI** | Sucidial ideation and attempt | 24.3 | 12.6 |
| 7 | Gizachew KD. et al, 2021 | Central Ethiopia | CS | 423 | **CIDI** | Sucidial ideation and attempt | 16 | 7.1 |

Note: **CIDI**: Composite International Diagnostic Interview, **CS:** Cross—sectional study

[11–14, 19–21]. Of all the included studies, 3 were conducted in Addis Ababa [12, 13, 19], 1 was conducted in Dessie [14], 1 was conducted at Harar [20], 1 was conducted at Debark [11], and the remaining one was conducted in central Ethiopia [21]. The study design of all studies was cross-sectional [11–14, 19–21].

In all of the included studies, the assessment instruments for suicidal ideation and attempts were CIDI [11–14, 19–21]. Besides, among the included studies, three used a sample size of more than 400 [13, 20, 21], and the remaining four [11, 12, 14, 19] used a sample size of less than 400. Moreover, considering the year of publication of the study, five were published in the past five years (2018 and above) [12, 14, 19–21], whereas the remaining twelve were published between 2013 and 2017 [11, 13] (Table 1).

## The pooled prevalence of suicidal ideation among adult HIV/AIDS patients in Ethiopia

Seven studies that assessed suicidal ideation and suicidal attempts in adult HIV/AIDS patients were included in the final meta-analysis to determine the pooled prevalence of suicidal ideation and suicidal attempts. The reported magnitude of suicidal ideation among the studies included in the current systematic review and meta-analysis varies from 9.4 in Dessie [14] to 33.6% in Debark [11]. The pooled prevalence of suicidal ideation among adult patients with HIV/AIDS in Ethiopia using the random effect model was 22.1% (95% CI 15.7–28.6). This pooled prevalence was under the influence of significant heterogeneity (I2 = 94.33%, p-value <0.001) from the variance between the included studies (Fig 2).

## Sub group analysis of the prevalence of suicidal ideation among adult HIV/AIDS patients in Ethiopia

Since the pooled prevalence of suicidal ideation was influenced by substantial heterogeneity, a subgroup analysis has been employed based on the study setting where the study was conducted, year of publication, and sample size < or >400. The pooled prevalence of suicidal ideation among adult HIV/AIDS patients in Addis Ababa administrative councils was 24.2% (95% CI 21.2, 27.2), with I2 = 26.28% and a p-value of 0.258.

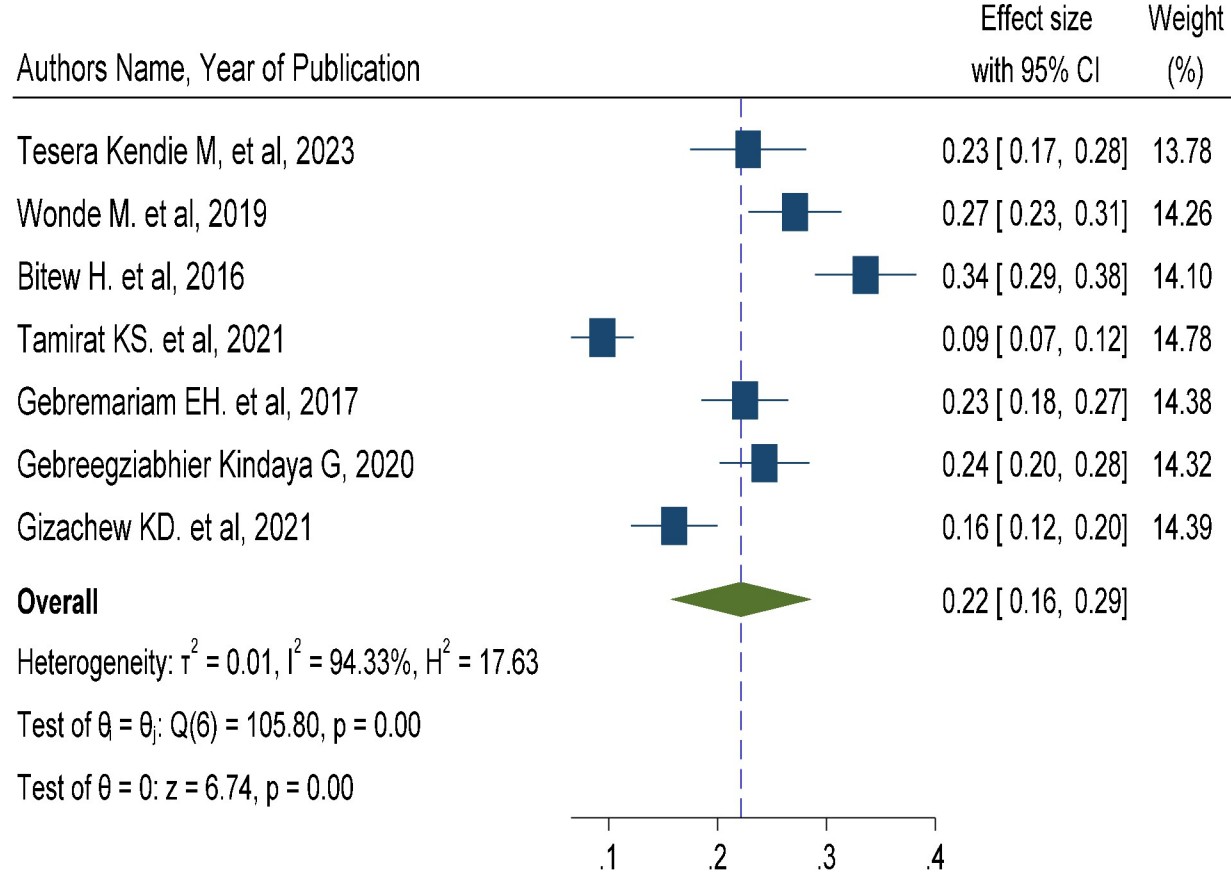

**Fig 2. A forest plot for the prevalence of suicidal ideation in HIV/AIDS patients.**

The pooled prevalence of suicidal ideation among HIV/AIDS patients in the city from Regions was 20.7% (95% CI 10.4, 31.0) with I2 = 96.55%, p-value <0.001.

Besides, a subgroup analysis considered the year of publication and sample size. The pooled prevalence of suicidal ideation was larger (28% (95% CI 17.1, 38.9) (I2 = 92%, P<0.001) in studies published before 2017 [11, 13] than in studies published after 2017 [12, 14, 23–25]; 19.8% (95% CI 12.6, 27) (I2 = 93.79%, P<0.001).

Moreover, the pooled prevalence of suicidal ideation was larger (24.5% (95% CI 21.9, 27.2) (I2 = 15.90%, p = 0.305) in studies that used a sample size≥400 [12, 13, 20, 21] than in studies that utilized a sample size<400 [11, 14, 19] (20.3% (95% CI 9.8, 30.9) (I2 = 96.25%, P<0.001) [Table 2].

## The pooled prevalence of suicidal attempt among adult HIV/AIDS patients in Ethiopia

Data regarding suicidal attempts were reported in all seven studies [11–14, 19–21]. The prevalence of suicidal attempts reported in these included studies ranges from 7.1 in Central Ethiopia [21] to 20.1% in Debark [11]. The pooled prevalence of suicidal attempts among adult patients with HIV/AIDS in Ethiopia using the random effect model was 12.4% (95% CI, 7.5,

**Table 2. A subgroup analysis of the prevalence of suicidal ideation among adult HIV/AIDS patients in Ethiopia.**

| Subgroup | Number of studies | Estimates | | Heterogeneity | | |
|---|---|---|---|---|---|---|
| | | Prevalence (%) | 95% CI | I2 (%) | Q(DF) | P-value |
| **Region** | | | | | | |
| City from Regions | 4 | 20.7 | 10.4, 31.0 | 96.55 | 86.94 (3) | <0.001 |
| Administrative city | 3 | 24.2 | 21.2, 27.2 | 26.28 | 2.71 (2) | 0.258 |
| **Sample size** | | | | | | |
| < 400 | 4 | 20.3 | 9.8, 30.9 | 96.25 | 80.10 (3) | <0.001 |
| ≥ 400 | 3 | 24.5 | 21.9, 27.2 | 15.90 | 2.38 (2) | 0.305 |
| **Year of publication** | | | | | | |
| From 2018–2023 | 5 | 19.8 | 12.6, 27 | 93.79 | 64.37 (4) | <0.001 |
| From 2013–2017 | 2 | 28 | 17.1, 38.9 | 92 | 12.50 (1) | <0.001 |

CI: confdence interval; DF: degree of freedom

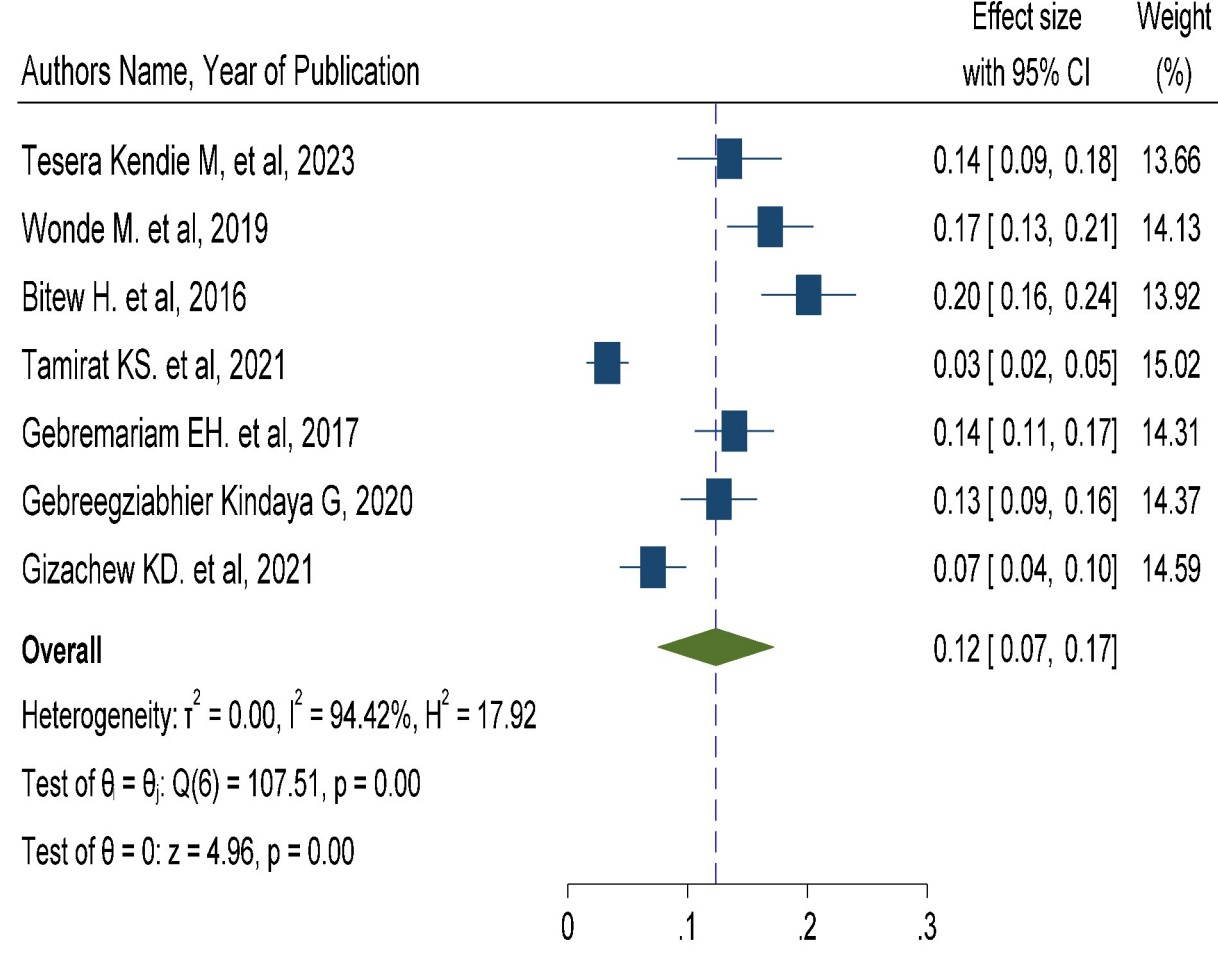

**Fig 3. A forest plot for the prevalence of suicidal attempt in HIV/AIDS patients.**

17.2). This pooled prevalence was under the influence of a significant heterogeneity (I2 = 94.42%, p-value <0.001) in the variance between the included studies (Fig 3).

## Sub group analysis of the prevalence of suicidal Attempt among adult HIV/AIDS patients in Ethiopia

Since the pooled prevalence of suicidal ideation was influenced by substantial heterogeneity, a subgroup analysis has been employed based on the study setting where the study was conducted, year of publication, and sample size < or >400. The pooled prevalence of suicidal attempts among adult HIV/AIDS patients in Addis Ababa administrative councils was 14.8% (95% CI 12.7, 17.0), with I2 = 0.00% and a p-value of 0.383.

The pooled prevalence of suicidal attempts among HIV/AIDS patients in the city from the regions was 10.6% (95% CI, 3.9, 17.3), with I2 = 95.74% and a p-value <0.001.

Besides, a subgroup analysis considered the year of publication and sample size. The pooled prevalence of suicidal attempts was larger (16.9% (95% CI 10.8, 23) (I2 = 81.9%, p = 0.019) in studies published before 2017 [11, 13] than in studies published after 2017 [12, 14, 23–25]: 10.5% (95% CI 5.3, 15.8) (I2 = 93.83%, P<0.001).

Moreover, the pooled prevalence of suicidal attempts was larger: 14.4% (95% CI 11.9, 16.8) (I2 = 36.00%, p = 0.210) in studies that used a sample size≥400 [12, 13, 20, 21] than studies that utilized a sample size<400 [11, 14, 19]; 10.8% (95% CI 3.8, 17.9) (I2 = 95.56%, P<0.001) [Table 3].

## Sensetivity analysis

To identify the source of heterogeneity that affects the pooled prevalence of suicidal ideation in adult HIV/AIDS patients, we conducted a sensitivity analysis. According to the findings of the sensitivity analysis, the pooled estimated prevalence of suicidal ideation obtained when every single study was excluded from the analysis was within the 95% confidence interval of the pooled prevalence of suicidal ideation when all studies were fitted together. Furthermore, the sensitivity analysis result showed that the pooled prevalence of suicidal ideation ranges between 20.58 (95% CI 15.76, 25.71) and 23.85% (95% CI 23.78, 23.92) when each study was left out of the analysis (Table 4).

Also, we did a sensitivity analysis for the prevalence of suicidal attempts, and the result showed that the pooled prevalence of suicidal attempts ranges between 11.08 (95% CI 6.35, 15.82) and 13.91% (95% CI 10.19, 17.62) when each study was left out of the analysis (Table 5).

**Table 3. A subgroup analysis of the prevalence of suicidal attempt among adult HIV/AIDS patients in Ethiopia.**

| Subgroup | Number of studies | Estimates | | Heterogeneity | | |
|---|---|---|---|---|---|---|
| | | Prevalence (%) | 95% CI | I2 (%) | Q(DF) | P-value |
| **Region** | | | | | | |
| City from Regions | 4 | 10.6 | 3.9, 17.3 | 95.74 | 70.43 (3) | <0.001 |
| Administrative city | 3 | 14.8 | 12.7, 17.0 | 0.00 | 1.92 (2) | 0.383 |
| **Sample size** | | | | | | |
| < 400 | 4 | 10.8 | 3.8, 17.9 | 95.56 | 67.52 (3) | <0.001 |
| ≥ 400 | 3 | 14.4 | 11.9, 16.8 | 36.00 | 3.12 (2) | 0.210 |
| **Year of publication** | | | | | | |
| From 2018–2023 | 5 | 10.5 | 5.3, 15.8 | 93.83 | 64.84 (4) | <0.001 |
| From 2013–2017 | 2 | 16.9 | 10.8, 23 | 81.9 | 5.53 (1) | 0.019 |

CI: confdence interval; DF: degree of freedom

**Table 4. A sensitivity analysis of the prevalence of suicidal ideation among adult HIV/AIDS patients in Ethiopia when each indicated studies are removed at a time with its 95% confidence interval.**

| No | Excluded study | Prevalence of sucidial ideation(%) | 95% CI |
|---|---|---|---|
| 1 | Tesera Kendie M, et al, 2023 | 22.06 | 14.76, 29.35 |
| 2 | Wonde M. et al, 2019 | 21.33 | 14.21, 28.45 |
| 3 | Bitew H. et al, 2016 | 20.24 | 14.19, 26.29 |
| 4 | Tamirat KS. et al, 2021 | 24.33 | 19.65, 29.01 |
| 5 | Gebremariam EH. et al, 2017 | 22.11 | 14.52, 29.70 |
| 6 | Gebreegziabhier Kindaya G, 2020 | 21.8 | 14.37, 29.23 |
| 7 | Gizachew KD. et al, 2021 | 23 .19 | 15.69, 30.69 |

CI: Confidence Interval

## Publication bias

The presence or absence of publication bias in the prevalence of suicidal ideation and attempts was checked with two methods. The first was Egger's publication bias plot. The result from this showed that publication bias exists and that its p-value is significant (P-value = 0.0172) for suicidal ideation and (P-value <0.001) for suicidal attempts, implying that there is significant publication bias for the prevalence of suicidal ideation and attempts in Ethiopia. Moreover, a visual inspection of a funnel plot for a Logit event rate of prevalence of suicidal ideation (Fig 4) and attempt (Fig 5) in adult HIV/AIDS patients against its standard error suggests supportive evidence for the presence of publication bias. To treat publication bias, we run a trim-and-fill analysis of publication bias, and we have one imputed study for both suicidal ideation and suicidal attempts. Finally, the adjusted pool prevalence of suicidal ideation is 20.3 with a 95% CI of 14–26.5, and the adjusted pooled prevalence of suicidal attempts is 11.1 with a 95% CI of 6.6–15.5.

## Factors associated with suicidal ideation in adult HIV/AIDS patients in Ethiopia

All seven studies reported the factors associated with suicidal ideation [11–14, 23–25]. The pooled effect of three studies [19–21] showed that adult people with HIV/AIDSwho were living alone were 4.98 times(AOR 4.98; 95% CI: 2.96–8.37) more likely to have suicidal ideation in Ethiopia as compared to adult patients with HIV/AIDS who were living with their family. The pooled effect of two studies [11, 19] showed that adult people with HIV/AIDS who had

**Table 5. A sensitivity analysis of the prevalence of suicidal attempt among adult HIV/AIDS patients in Ethiopia when each indicated studies are removed at a time with its 95% confidence interval.**

| No | Excluded study | Prevalence of sucidial ideation(%) | 95% CI |
|---|---|---|---|
| 1 | Tesera Kendie M, et al, 2023 | 12.19 | 6.75, 17.63 |
| 2 | Wonde M. et al, 2019 | 11.61 | 6.45, 16.76 |
| 3 | Bitew H. et al, 2016 | 11.08 | 6.35, 15.82 |
| 4 | Tamirat KS. et al, 2021 | 13.91 | 10.19, 17.62 |
| 5 | Gebremariam EH. et al, 2017 | 12.12 | 6.59, 17.64 |
| 6 | Gebreegziabhier Kindaya G, 2020 | 12.34 | 6.68, 17.99 |
| 7 | Gizachew KD. et al, 2021 | 13.28 | 7.42, 19.15 |

CI: confidence interval

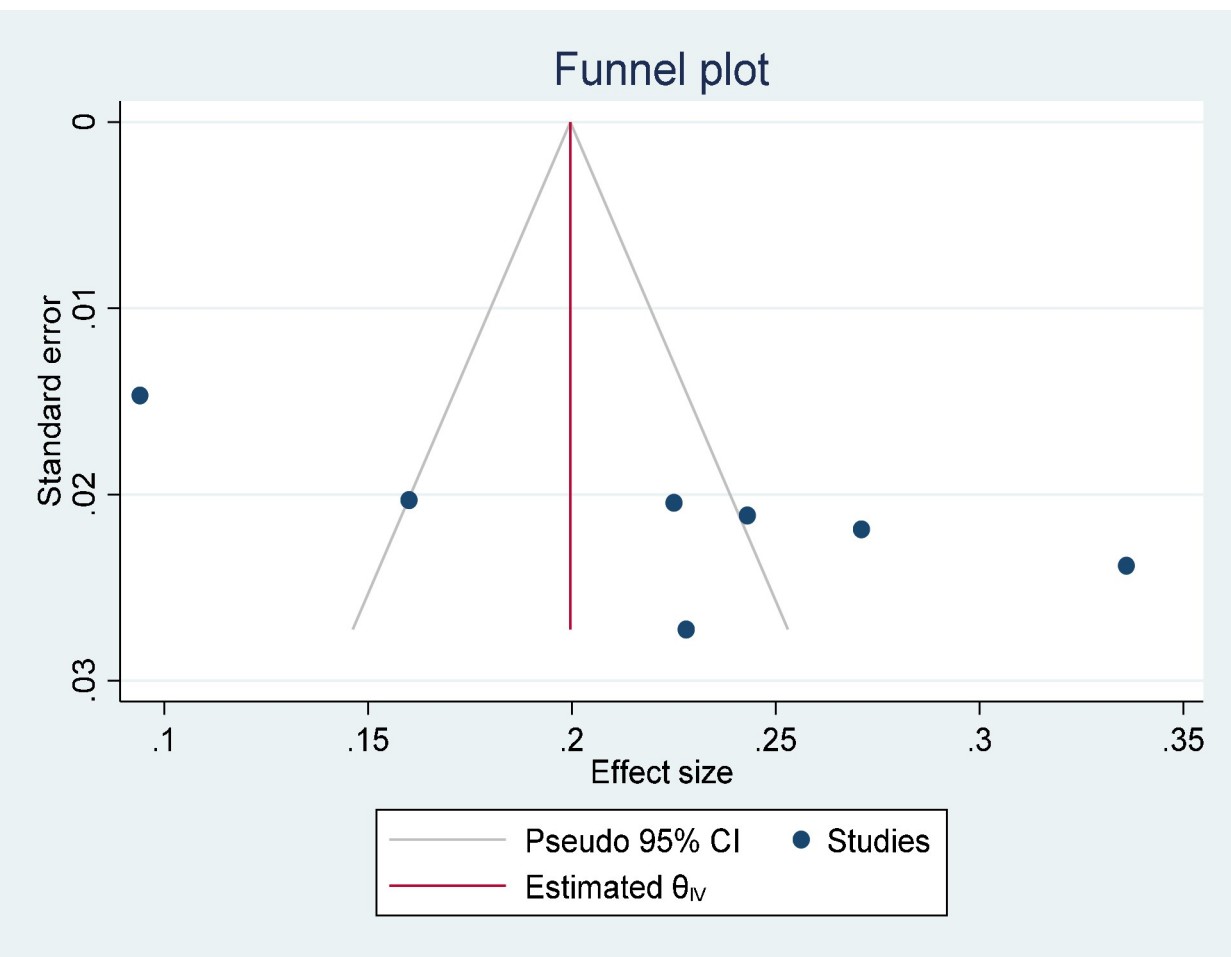

**Fig 4. A funnel plot for the prevalence of suicidal ideation in adult HIV/ AIDS patients.**

comorbidity or other opportunistic infection were 4.67 times (AOR 4.67; 95% CI: 2.57–8.48) more likely to have suicidal ideation in Ethiopia as compared to adult people with HIV/AIDS who had no comorbidity or other opportunistic infection.

The pooled effect of two studies [12, 19] showed that adult people with HIV/AIDS who are female were 2.86 times (AOR 2.86; 95% CI: 1.76–4.62) more likely to have suicidal ideation in Ethiopia as compared to adult people with HIV/AIDS who are male.

The pooled effect of two studies [12, 13] showed that adult people with HIV/AIDS who had WHO clinical stage III of HIV were 3.69 times (AOR 3.69; 95% CI: 2.15, 6.32) more likely to have suicidal ideation in Ethiopia as compared to adult people with HIV/AIDS who had WHO clinical stage I of HIV (Table 6).

The pooled effect of two studies [12, 13] showed that adult people with HIV/AIDS who had WHO clinical stage IV of HIV were 5.43 times (AOR 5.43; 95% CI: 2.81, 10.53) more likely to have suicidal ideation in Ethiopia as compared to adult people with HIV/AIDS who had WHO clinical stage I of HIV.

The pooled effect of four studies [11–14] showed that adult people with HIV/AIDS who had co-morbid depression were 5.25 times (AOR 5.25; 95% CI: 4.05, 6.80) more likely to have suicidal ideation in Ethiopia as compared to adult people with HIV/AIDS who had no co-morbid depression (Table 6).

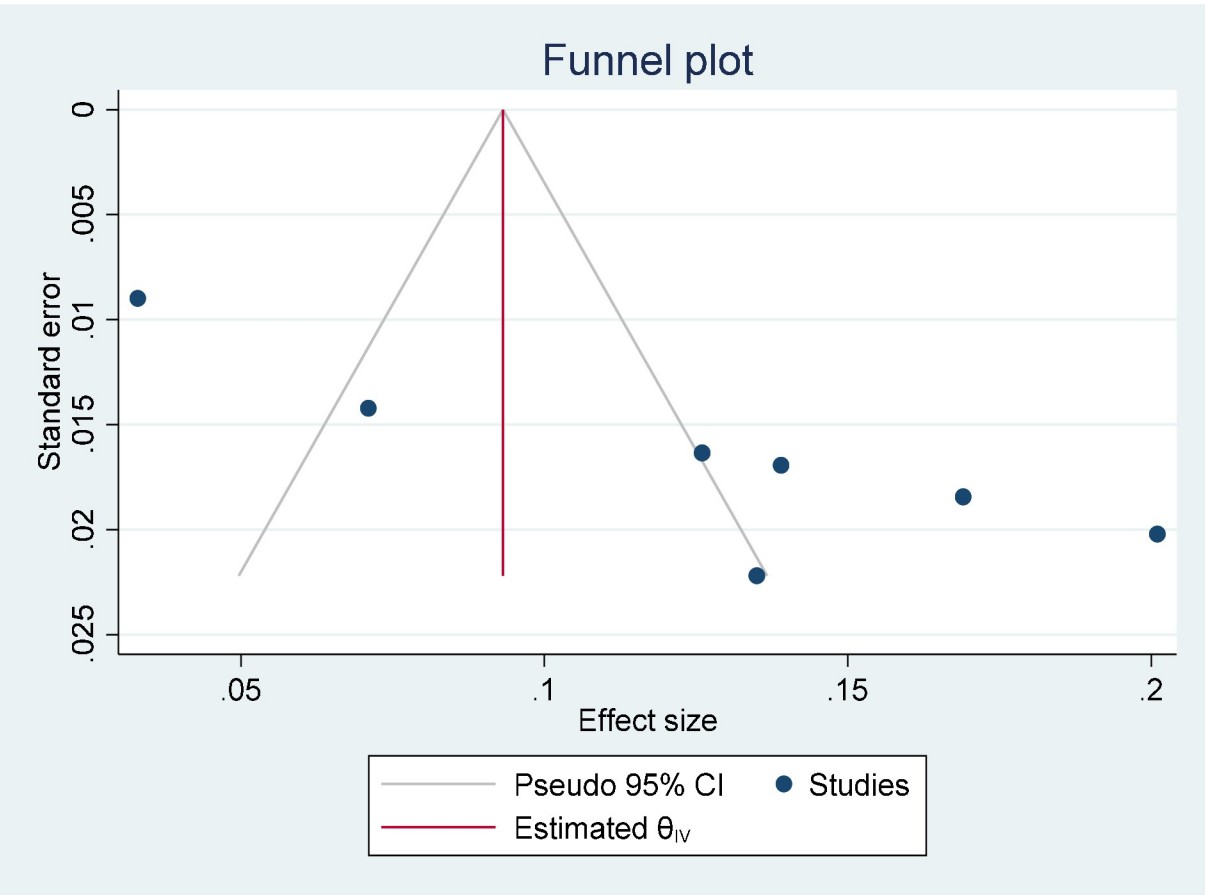

**Fig 5. A funnel plot for the prevalence of suicidal attempt in adult HIV/ AIDS patients.**

The pooled effect of two studies [12, 13] showed that adult people with HIV/AIDS who had perceived HIV stigma were 2.53 times (AOR 2.53; 95% CI: 1.67–3.84) more likely to have suicidal ideation in Ethiopia as compared to adult people with HIV/AIDS who had no perceived HIV stigma. The pooled effect of two studies [13, 21] showed that adult people with HIV/AIDS who had a family history of suicidal attempts were 2.79 times (AOR 2.79; 95% CI: 1.38–5.66) more likely to have suicidal ideation in Ethiopia as compared to adult people with HIV/AIDS who had no family history of suicidal attempts (Table 6).

**Table 6. Characteristics ofpooled associated factors with suicidal ideation among adult HIV/AIDS patients in Ethiopia.**

| Variables | AOR | 95% CI | P value |
|---|---|---|---|
| Living alone | 4.98 | 2.96, 8.37 | <0.001 |
| Having comorbidity or other opportunistic infection | 4.67 | 2.57, 8.48 | <0.001 |
| Female sex | 2.86 | 1.76, 4.62 | <0.001 |
| WHO clinical stage III of HIV | 3.69 | 2.15, 6.32 | <0.001 |
| WHO clinical stage IV of HIV | 5.43 | 2.81, 10.53 | <0.001 |
| Co-morbid depression | 5.25 | 4.05, 6.80 | <0.001 |
| Perceived HIV stigma | 2.53 | 1.67, 3.84 | <0.001 |
| Family'history of suicidal attempt | 2.79 | 1.38, 5.66 | 0.004 |

**Table 7. Characteristics of pooled associated factors with suicidal attempt among adult HIV/AIDS patients in Ethiopia.**

| Variables | AOR | 95% CI | P value |
|---|---|---|---|
| Female sex | 4.33 | 2.36, 7.96 | <0.001 |
| opportunistic infections | 2.73 | 1.69, 4.41 | <0.001 |
| WHO clinical stage III of HIV | 3.78 | 2.04, 7.03 | <0.001 |
| co-morbid depression | 3.47 | 2.38, 5.05 | <0.001 |
| poor social support | 3.02 | 1.78, 5.13 | <0.001 |
| WHO clinical stage IV | 7.39 | 3.54, 15.41 | <0.001 |

## Factors associated with suicidal attempt in adult HIV/AIDS patients in Ethiopia

Of all the included studies, six reported the factors associated with suicidal attempts [11–13, 19–21]. The pooled effect of two studies [11, 13] showed that adult people with HIV/AIDS who were female were 4.33 times (AOR 4.33; 95% CI: 2.36, 7.96) more likely to perform suicidal attempts in Ethiopia as compared to adult patients with HIV/AIDS who were male. The pooled effect of two studies [11, 12] showed that adult people with HIV/AIDS who had opportunistic infection were 2.73 times (AOR 2.73; 95% CI: 1.69, 4.41) more likely to perform suicidal attempts in Ethiopia as compared to adult people with HIV/AIDS who had not had opportunistic infection (Table 7).

The pooled effect of two studies [12, 13] showed that adult people with HIV/AIDS who had WHO clinical stage III of HIV were 3.78 times (AOR 3.78; 95% CI: 2.04, 7.03) more likely to perform suicidal attempt in Ethiopia as compared to adult people with HIV/AIDS who had WHO clinical stage I of HIV (Table 7). The pooled effect of four studies [12, 13, 19, 20] showed that adult people with HIV/AIDS who had co-morbid depression were 3.47 times (AOR 3.47; 95% CI: 2.38, 5.05) more likely to perform suicidal attempt in Ethiopia as compared to adult people with HIV/AIDS who had no co-morbid depression (Table 7).

The pooled effect of two studies [11, 12] showed that adult people with HIV/AIDS who had poor social support were 3.02 times (AOR 3.02; 95% CI: 1.78, 5.13) more likely to perform suicidal attempts in Ethiopia as compared to adult people with HIV/AIDS who had good social support. The pooled effect of two studies [13, 20] showed that adult people with HIV/AIDS who had WHO clinical stage IV were 7.39 times (AOR 7.39; 95% CI: 3.54, 15.41) more likely to perform suicidal attempts in Ethiopia as compared to adult people with HIV/AIDS who had WHO clinical stage I (Table 7).

## Discussion

In this systematic review and meta-analysis study, seven studies that were conducted between 2013 and 2023 with 2593 participants were analyzed to estimate suicidal ideation, suicidal attempts, and their associated factors among adult HIV/AIDS patients in Ethiopia. All seven studies assessed both suicidal ideation and suicidal attempts as primary outcome variables. According to this meta-analysis, the pooled prevalence of suicidal ideation and suicidal attempts was 22.1% and 11.1%, respectively. Living alone, having comorbidity or other opportunistic infection, having WHO clinical stage III of HIV, having WHO clinical stage IV of HIV, having co-morbid depression, having perceived HIV stigma, and having a family history of suicidal attempts were significantly associated with suicidal Ideation. Being female, having opportunistic infections, having WHO clinical stage IIIHIV, having co-morbid depression, having poor social support, and having WHO clinical stage IV were significantly associated with suicidal attempts.

The pooled prevalence of suicidal ideation among the included studies was 20.3, with a 95% CI of 14–26.5. This was in line with the result of a systematic review and meta-analysis study that reported that the prevalence of suicidal ideation was 22.3% among college students [22], 21.7% among patients with HIV/AIDS in Africa [6], and 24.38% among studies on HIV/AIDS patients worldwide [23].

However, this was higher than studies in Thailand, which were 15.5% [24], 14.0% in Canada [25], and 10% in the United States [26]. This difference might be due to socio-economic, socio-demographic, and cultural differences.

On the other hand, the pooled prevalence in the present study was lower than the results of studies in China: 31.6%, 64% [4, 27], and 41.6% among homeless people in Ethiopia [28]. This difference might be due to the fact that the prevalence report from Chinese studies was from single studies, but the current study is a meta-analysis study that might provide a more precise result than single studies. Besides this, the variation in studies among homeless people in Ethiopia is due to the fact that people who are homeless are among the most at-risk groups and experience severe psychosocial and financial issues, which may increase their chance of developing suicidal thoughts [29].

The prevalence of suicidal ideation showed variation based on the city of study, year of publication, and sample size of the study. Our subgroup analysis showed that the pooled prevalence of suicidal ideation among adult HIV/AIDS patients was 24.2% in Addis Ababa administrative councils, higher than the pooled prevalence of suicidal ideation in the city from the regions (Amhara and Harar) of 20.7%. This difference might be due to variations in the level of urbanization. When we compare cities found in the region and Addis Ababa administrative councils, the level of urbanization is high in Addis Ababa. Numerous issues may arise as a result of urbanisation, including high population density, concentrated poverty, alienation from nature, increased noise and air pollution, and social isolation [30]. Common mental disorders are being diagnosed more frequently, and self-harm and suicide are becoming more common, all of which are related to the general trend towards urban living [31].

The pooled prevalence of suicidal attempts in our study was 11.1 with a 95% CI of 6.6–15.5. This finding is in line with the systematic review and meta-analysis study: 13.08% worldwide [10], 9% in Japan [32], and 8.2% in Thailand [24].

However, this result was higher than the results in studies in Nigeria: 1.3% [9] and 3.5% in Canada [33]. In contrast, the result of this study was lower than the studies among homeless people in Ethiopia, 28.80% [28], 23% in France [9], and 22.6% in China [4]. The possible explanation of this variation might be due to the difference in disclosure status of suicidal attempts attributed to environmental and socio-economic factors between the above-mentioned studies and the Ethiopian studies included in this analysis. Besides this, the presented study is a systematic review and meta-analysis study, and the comparator studies such as China [4] and France [33] are single studies that might also cause variation in the magnitude of suicidal attempts. Moreover, the variation from studies among homeless people in Ethiopia is due to the fact that People who are homeless are among the most at-risk groups and experience severe psychosocial and financial issues, which may increase their chance of developing suicidal attempts [29].

The prevalence of suicidal attempts showed variation based on the city of study, year of publication, and sample size of the study. Our subgroup analysis showed that the pooled prevalence of suicidal attempts among adult HIV/AIDS patients was 14.8% in Addis Ababa administrative councils, higher than in the city from Regions (Amhara and Harar) at 10.6%. The reasons discussed for suicidal ideation, could also be applicable to suicidal attempts.

Advanced stages of AIDS, such as WHO clinical stage III and WHO clinical stage IV, were factors associated with suicidal ideation [12, 13] and attempts [12, 13, 20]. This could be

because the patient's symptoms are getting worse and their clinical conditions are deteriorating, which might also give them a sense of hopelessness and make them more suicidal.

Having opportunistic infections [11–13] was associated with suicidal ideation [19] and attempts [11, 12]. This might be a result of the rise in AIDS-defining opportunistic illnesses, which increase patient burdens and lower their quality of life.

Co-morbid depression [11–14] was also associated with suicidal ideation and attempts [13, 19, 20] in adult HIV/AIDS patients in Ethiopia. This finding was supported by a study in China [4]. The reason for this is that depression will deplete the involvement of the monoaminergic neurotransmitter systems, particularly the serotonergic ones, in our brain, and studies showed that a decrease in serotonin would have an impact on suicidal ideation and attempts [34, 35]. Additionally, the direct social effects of depression, such as social withdrawal, hopelessness, and worthlessness, may be responsible for suicidal ideation and attempts.

In this study, poor social support was found to be a factor associated with suicidal ideation [11, 14] and attempts [11, 12]. This finding was supported by a study in France [33]. This may be because HIV/AIDS patients who have poor social support find it difficult to cope with their chronic illness and the psychological burden it imposes on them on their own and feel alienated to the point where their risk of committing suicide rises [36].

Furthermore, disclosure status [19], living arrangement [19], being single [11], not on HAART [13], low monthly income [21], being female [11, 12], perceived HIV stigma [11, 12] Experiencing mild and moderate-to-severe depression and anxiety symptoms [21] and being gossiped about sometimes in the last 12 months of the study period due to HIV status [21] were the factors associated with suicidal ideation and suicidal attempts in adult patients with HIV/AIDS in Ethiopia.

## Strengths and limitations of the study

There are strengths and limitations to this systematic review and meta-analysis study. The employment of a predetermined search method that reduces the reviewer's bias is the first strength of the study. The study's quality evaluation and data extraction were carried out by independent reviewers, which further reduced reviewer bias. This was the study's second strength. A strength was the use of sensitivity analysis and subgroup analysis to find the source of heterogeneity. Contrarily, the heterogeneity in the study that could skew the results' interpretation is what gives birth to its limitations. Another drawback is that the validity of the estimate may be reduced by the subgroup analysis using only a small number of studies.

## Conclusion

This systematic review and meta-analysis study revealed that the adjusted pooled prevalence of suicidal ideation and attempt among HIV/AIDS patients was high in Ethiopia. Living alone, having comorbidity or other opportunistic infections, WHO clinical stage III of HIV, WHO clinical stage IV of HIV, co-morbid depression, perceived HIV stigma and a family history of suicidal attempts are significantly associated with suicidal ideation.

Being female, having opportunistic infections, having WHO clinical stage III of HIV, having WHO clinical stage IV of co-morbid depression, and having poor social support are significantly associated with suicidal attempts.

Hence, to reduce suicidal ideation and attempts among people living with HIV, much consideration has to be given to modifying the associated factors by strengthening routine antiretroviral therapy and integrating it with mental health services. Further qualitative studies are suggested to incorporate the lived experience of people living with HIV in the context of suicidal ideation and attempts.

### Implications of the results for practice, policy, and future research

The results of this systematic review and meta-analysis on suicidal ideation and attempt on HIV/AIDS patients have potential implications for for practice, policy, and future research.

**For practitioner.** The finding of this study would be important evidence for clinical practitioners working on ART clinic to give holistic service by giving much emphasis on mental illness including sucidial ideation and attempt, since suicidal ideation and attempt are not given much attention.

**For policy maker.** The policymakers and program planners would design an integrated holistic approach in the management of people living with HIV/AIDS by considering mental health as a priority.

**For future researcher.** The high pooled magnitude of suicidal ideation and attempt in the target population reported in this study relative to the general population will be a motive for further research to explore lived experience of people living with HIV in the sucidial ideation and attempt context.

## Acknowledgments

We authors of this research work want to acknowledge the authors of the included studies as they are the basis for our investigation.

## Author Contributions

**Conceptualization:** Eyob Ketema Bogale, Tadele Derbew Kassie, Tadele Fentabil Anagaw, Elyas Melaku Mazengia, Sintayehu Shiferaw Gelaw, Eneyew Talie Fenta, Habitu Birhan Eshetu, Natnael Kebede.

**Data curation:** Eyob Ketema Bogale.

**Formal analysis:** Eyob Ketema Bogale.

**Investigation:** Eyob Ketema Bogale.

**Methodology:** Eyob Ketema Bogale, Amare Zewdie, Tadele Derbew Kassie, Tadele Fentabil Anagaw, Elyas Melaku Mazengia, Sintayehu Shiferaw Gelaw, Eneyew Talie Fenta, Habitu Birhan Eshetu, Natnael Kebede.

**Project administration:** Eyob Ketema Bogale.

**Resources:** Eyob Ketema Bogale.

**Software:** Eyob Ketema Bogale, Amare Zewdie, Tadele Derbew Kassie, Tadele Fentabil Anagaw, Elyas Melaku Mazengia, Eneyew Talie Fenta, Habitu Birhan Eshetu.

**Supervision:** Eyob Ketema Bogale.

**Validation:** Eyob Ketema Bogale.

**Visualization:** Eyob Ketema Bogale.

**Writing – original draft:** Eyob Ketema Bogale, Amare Zewdie, Tadele Derbew Kassie, Tadele Fentabil Anagaw, Elyas Melaku Mazengia, Sintayehu Shiferaw Gelaw, Eneyew Talie Fenta, Habitu Birhan Eshetu, Natnael Kebede.

**Writing – review & editing:** Eyob Ketema Bogale, Tadele Derbew Kassie, Tadele Fentabil Anagaw, Elyas Melaku Mazengia, Sintayehu Shiferaw Gelaw, Eneyew Talie Fenta, Habitu Birhan Eshetu, Natnael Kebede.

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
