## [Decision Letter · Decision Letter 0]

18 Oct 2023

PONE-D-23-21279Suicidal ideation, attempt, and its associated factors among Adult HIV/AIDS patients in Ethiopia: a systematic review and meta-analysis studyPLOS ONE

Dear Dr. Bogale,

Thank you for submitting your manuscript to PLOS ONE. After careful consideration, we feel that it has merit but does not fully meet PLOS ONE’s publication criteria as it currently stands. Therefore, we invite you to submit a revised version of the manuscript that addresses the points raised during the review process.

We look forward to receiving your revised manuscript.

Kind regards,

Chalachew Kassaw Demoze, Msc

Academic Editor

PLOS ONE

Journal Requirements:

Additional Editor Comments:

Dear Author , It is a good work

Here are some of the concerns about a paper

1. first it is better to re-write a manuscriput with a correct alpahabetical and grammatical revisions

2. it is better to state a review artclicles on the introduction part of the manuscript , I mean you only stated a primary study

3. It is better not to merege sub group analysis (Amhara vs Harar) , rather it is better to do regional vs adminstarative city analaysis

4. On the discussion part of the manuscrput , please discuss with review articles done between 2013-2023.

Reviewers' comments:

Reviewer's Responses to Questions

**Comments to the Author**

1. Is the manuscript technically sound, and do the data support the conclusions?

Reviewer #1: Yes

2. Has the statistical analysis been performed appropriately and rigorously? 

Reviewer #1: Yes

3. Have the authors made all data underlying the findings in their manuscript fully available?

Reviewer #1: Yes

4. Is the manuscript presented in an intelligible fashion and written in standard English?

Reviewer #1: Yes

5. Review Comments to the Author

Reviewer #1: Dear Editor,

Thank you for the opportunity to review the manuscript entitled “Suicidal ideation, attempt, and its associated factors among Adult HIV/AIDS patients in Ethiopia: a systematic review and meta-analysis study”. The manuscript is well written. Yet, there are some points the authors should consider and I have tried to put the points section by section.

Dear Authors,

Thank you for your contribution in the field. I have got the opportunity to review your manuscript and found it well written. However, there are some points you need to consider, and I have mentioned them section by section as follows:

Abstract:

Background: The background section of the abstract should focus on stating the problem than defining suicidal attempt and ideation which can be done under the main text. It is good to clearly state the aim /objective of the review.

Results: The result section should highlight the main findings in line with the objective of the study. Accordingly, factors associated with suicidal ideation and/or attempt should be discussed. In addition, detail description of the area specific pooled prevalence might not be very important under the abstract. Rather, you can describe if there is significant geographical heterogeneity.

Conclusion: “The pooled magnitude of suicidal ideation and attempt was high, and factors like advanced stages of AIDS, co-morbid depression, perceived stigma, and poor social support were related to it.” From this statement it is not known whether the factors are associated with suicidal ideation or attempt. Thus, please, make it specific.

Introduction:

The introduction is well framed. Yet, there are some points you need to consider.

1. Your population is PLWIHV. Thus, detailed statement of the problem regarding the general population would not have much significance. Accordingly, please highlight the burden in the general briefly and focus on the burden among HIV people.

Methods:

Data extraction: Was the data extraction done independently by the mentioned authors? What was the remedy if there is a discrepancy between the extracted data? Similarly, for the quality assessment how was it performed?

Results:

Identification of studies: On the PRISMA flowchart, please identify the number of articles identified from each data base. In addition, include the reason of exclusion for the 287 articles in the chart.

It is good to do the subgroup analysis. However, how did you treat the studies conducted in Amhara and Harar together? This is because the heterogeneity with regard to the suicidal ideation still exists when the studies are combined together or how do you explain the persistent heterogeneity?

Please, replace P=0.0000 by P<0.001.

Please, avoid repetitions; for example, the following statements were repeated a number of times….. “The seven studies included in the meta-analysis, three were from Addis Ababa administrative councils [11, 12, 18], and four were from cities found in two regions, Amhara and Harar [10, 13, 19, 20].”

For the factors associated with the suicidal ideation and attempt, it is good to just describe the pooled effect sizes than the study specific effect sizes.

Discussion:

The first paragraph of a discussion should be the highlight of the current study and the main findings. Thus please, describe the highlight of your study and the main findings and then discuss each finding one by one.

Please, make sure that you compare your finding with comparable studies i.e., systematic reviews and meta-analyses.

….. “Our subgroup analysis showed that the pooled prevalence of suicidal ideation among adult HIV/AIDS patients was 24.2% in Addis Ababa administrative councils, higher than the pooled prevalence of suicidal ideation in the city from the regions (Amhara and Harar) of 20.7%. This difference might be due to variations in the level of urbanization.” Please, explain how urbanization is related to suicidal ideation.

Conclusion:

Conclusion is about the implication of your study than the mere figure of the results. Accordingly, please, describe the implication of the pooled prevalence of suicidal ideation and attempt than the proportion itself.

6. PLOS authors have the option to publish the peer review history of their article (what does this mean?). If published, this will include your full peer review and any attached files.

Reviewer #1: **Yes: **Leta Adugna Geleta

---

## [Author Response · Author response to Decision Letter 0]

21 Oct 2023

RESPONSE TO EDITOR AND REVIEWERS

RESPONSE TO EDITOR

RESPONSE: Thank you for coordinating the review process and fruitful comments. We have revised the manuscript and addressed Reviewer’s comments.

Journal Requirements: 

COMMENT: Journal Requirements:

RESPONSE: We have checked and attest that all formatting and style requirements have been met and revised based on the guideline 

COMMENT: 2. Please review your reference list to ensure that it is complete and correct. If you have cited papers that have been retracted, please include the rationale for doing so in the manuscript text, or remove these references and replace them with relevant current references. Any changes to the reference list should be mentioned in the rebuttal letter that accompanies your revised manuscript. If you need to cite a retracted article, indicate the article’s retracted status in the References list and also include a citation and full reference for the retraction notice.

RESPONSE: Thank you for fruitful comments. We have reviewed our reference list and we got it complete and correct.

COMMENT: Additional Editor Comments:Dear Author , It is a good work

Here are some of the concerns about a paper

1. first it is better to re-write a manuscriput with a correct alpahabetical and grammatical revisions

RESPONSE: Thank you for your kind word and fruitful comments. We have made extensive revision in our manuscript to reduce alpahabetical and grammatical errors.

 COMMENT: 2. it is better to state a review artclicles on the introduction part of the manuscript , I mean you only stated a primary study

RESPONSE: Thank you for your kind word and fruitful comments. We have revised it by adding results of study from systematic review and meta analysis.

COMMENT: 3. It is better not to merege sub group analysis (Amhara vs Harar) , rather it is better to do regional vs adminstarative city analaysis

RESPONSE: Thank you for your kind word and fruitful comments. We have seen it and when we see our data we have 3 studies from Amhara Region and only 1 study from Harer. If we classify our subgroup analysis in to 3 such as adminstarative city analaysis, Amhara region and Harer region, we can not address the pooled prevalence in Harer Region since it has a single study. So That, we try to categorize studies in to two based on their homogeneous characteristics and we assumed that all cities from each region in Ethiopia have comparable urbanization and infracture. So that it is better to categorize it in two as adminstarative city and city from Regions. I kindly ask you to consider this during review.

COMMENT: 4. On the discussion part of the manuscrput , please discuss with review articles done between 2013-2023. 

RESPONSE: Thank you for your kind word and fruitful comments. We have revised it based on your comment.

RESPONSE TO REVIEWER 1:

Dear Authors,

Thank you for your contribution in the field. I have got the opportunity to review your manuscript and found it well written. However, there are some points you need to consider, and I have mentioned them section by section as follows:

RESPONSE: We thank the reviewer for kind words. We have revised our manuscript based on the comments as described below.

Abstract:

REVIEWER COMMENT:  Background: The background section of the abstract should focus on stating the problem than defining suicidal attempt and ideation which can be done under the main text. It is good to clearly state the aim /objective of the review.

RESPONSE: We thank the reviewer for fruitful comments. We have revised our manuscript based on the comments by deleting definition of suicidal attempt and ideation , and focusing on stating the problem and adding sentence that describe the aim /objective of the review. 

REVIEWER COMMENT: Results: The result section should highlight the main findings in line with the objective of the study. Accordingly, factors associated with suicidal ideation and/or attempt should be discussed. In addition, detail description of the area specific pooled prevalence might not be very important under the abstract. Rather, you can describe if there is significant geographical heterogeneity.

RESPONSE: Thank you for fruitful comments. We have revised it by deleting unnecessary detail and by highlighting main findings in line with the objective of the study including factors associated with suicidal ideation and/or attempt.

REVIEWER COMMENT: Conclusion: “The pooled magnitude of suicidal ideation and attempt was high, and factors like advanced stages of AIDS, co-morbid depression, perceived stigma, and poor social support were related to it.” From this statement it is not known whether the factors are associated with suicidal ideation or attempt. Thus, please, make it specific. 

RESPONSE: Thank you for fruitful comments. We have revised it to make it clear. 

Introduction:

REVIEWER COMMENT: The introduction is well framed. Yet, there are some points you need to consider. 1. Your population is PLWIHV. Thus, detailed statement of the problem regarding the general population would not have much significance. Accordingly, please highlight the burden in the general briefly and focus on the burden among HIV people.

RESPONSE: thank you for fruitful comments. We have revised it based on your comments.

Methods:

REVIEWER COMMENT: Data extraction: Was the data extraction done independently by the mentioned authors? What was the remedy if there is a discrepancy between the extracted data? 

RESPONSE: thank you for fruitful comments. We have take the following remedy if a discrepancy between the extracted data were occurred:- During the extraction process, discrepancy between the extracted data were resolved through logical discussion among the three authors, and the final consensus was approved with the participation of authors (NK and HBE).

REVIEWER COMMENT: Similarly, for the quality assessment how was it performed?

RESPONSE: Thank you for fruitful comments. The quality assessment was performed as follow:-

The Newcastle-Ottawa quality assessment scale [16] was used to rate the caliber of the research that was considered for the final analysis. This scale allows for possible scores ranging from 0 to 10. On this scale, a score of 8 or more indicated good quality, a score of 3 to 7 indicated moderate quality, and a score of less than this indicated low quality. The quality of the included studies were evaluated by two independent authors (EKB and TF) using the Newcastle-Ottawa quality assessment scale [16]. The third author (NK) discussed and resolved any disagreements. Out of the seven studies that were considered, two had moderate quality and five had good quality.

Results:

REVIEWER COMMENT: Identification of studies: On the PRISMA flowchart, please identify the number of articles identified from each data base. In addition, include the reason of exclusion for the 287 articles in the chart.

RESPONSE: Thank you for fruitful comments. We have revised it based on your comments. We have included the number of articles identified from each data base and we have provide reasons of exclusion for the 287 articles in the chart.

REVIEWER COMMENT: It is good to do the subgroup analysis. However, how did you treat the studies conducted in Amhara and Harar together? This is because the heterogeneity with regard to the suicidal ideation still exists when the studies are combined together or how do you explain the persistent heterogeneity?

RESPONSE: Thank you for your kind word and fruitful comments. We have seen it and when we see our data we have 3 studies from Amhara Region and only 1 study from Harer. If we classify our subgroup analysis in to 3 such as adminstarative city analaysis, Amhara region and Harer region, we can not address the pooled prevalence in Harer Region since it has a single study. So That, we try to categorize studies in to two based on their homogeneous characteristics and we assumed that all cities from each region in Ethiopia have comparable urbanization and infrascture. So that it is better to categorize it in two as adminstarative city and city from Regions. We have checked that the source of heterogeneity with regard to the suicidal ideation still exists due to the studies are combined together or not, by re analysing our data by excluding Harer region from the group and we got I2 (%)= 97.33. So that, please try to noticed that persistent heterogeneity in this study not come from combining two regions in one category. It might be due to unexplained other reasons. I kindly ask you to consider this during review.

REVIEWER COMMENT: - replace P=0.0000 by P<0.001 Please, replace P=0.0000 by P<0.001.

RESPONSE: Thank you for fruitful comments. We have revised it by replacing P=0.0000 by P<0.001

REVIEWER COMMENT:- Please, avoid repetitions; for example, the following statements were repeated a number of times….. “The seven studies included in the meta-analysis, three were from Addis Ababa administrative councils [11, 12, 18], and four were from cities found in two regions, Amhara and Harar [10, 13, 19, 20].”

RESPONSE: Thank you for fruitful comments. We have revised it by deleting redundant paragraph as mentioned above.

REVIEWER COMMENT: For the factors associated with the suicidal ideation and attempt, it is good to just describe the pooled effect sizes than the study specific effect sizes.

RESPONSE: Thank you for fruitful comments. We have revised our manuscript by describing the pooled effect sizes rather than the study specific effect sizes. 

Discussion:

REVIEWER COMMENT: The first paragraph of a discussion should be the highlight of the current study and the main findings. Thus please, describe the highlight of your study and the main findings and then discuss each finding one by one. Please, make sure that you compare your finding with comparable studies i.e., systematic reviews and meta-analyses.

RESPONSE: Thank you for fruitful comments. We have revised it based on your comment by focusing highlighting of the current study and the main findings as first paragraph of a discussion as following paragraph and then we discussed each finding one by one. Moreover, we make sure for you that we compared our finding with other comparable studies such as systematic reviews and meta-analyses.

In this systematic review and meta-analysis study, seven studies that were conducted between 2013 and 2023 with 2593 participants were analyzed to estimate suicidal ideation, suicidal attempts, and its associated factors among adult HIV/AIDS patients in Ethiopia. All seven studies assessed both suicidal ideation and suicidal attempts as a primary outcome variable. According to this metaanalysis, the pooled prevalence of suicidal ideation and suicidal attempt was 22.1%, and 11.1, respectively. Living alone, having comorbidity or other opportunistic infection, having WHO clinical stage III of HIV, having WHO clinical stage IV of HIV, having co-morbid depression, having perceived HIV stigma and having family history of suicidal attempt were significantly associated with suicidal Ideation. Being female, having opportunistic infections, WHO clinical stage III of HIV, co-morbid depression, having poor social support, and having WHO clinical stage IV were significantly associated with suicidal attempt.

REVIEWER COMMENT:….. “Our subgroup analysis showed that the pooled prevalence of suicidal ideation among adult HIV/AIDS patients was 24.2% in Addis Ababa administrative councils, higher than the pooled prevalence of suicidal ideation in the city from the regions (Amhara and Harar) of 20.7%. This difference might be due to variations in the level of urbanization.” Please, explain how urbanization is related to suicidal ideation.

RESPONSE: Thank you for fruitful comments. In our manuscript, we provide explanation for how urbanization is related to suicidal ideation as follow:-

When we compare cities found in the region and Addis Ababa administrative councils, the level of urbanization is high in Addis Ababa. Numerous issues may arise as a result of urbanisation, including high population density, concentrated poverty, alienation from nature, increased noise and air pollution, and social isolation [31]. Common mental disorders are being diagnosed more frequently, and self-harm and suicide are becoming more common, all of which are related to the general trend towards urban living (32).

Conclusion:

REVIEWER COMMENT: Conclusion is about the implication of your study than the mere figure of the results. Accordingly, please, describe the implication of the pooled prevalence of suicidal ideation and attempt than the proportion itself. 

RESPONSE: We thank the reviewer for kind words and fruitful comments. We have revised it by describing the implication of the pooled prevalence of suicidal ideation and attempt than the proportion itself.

---

## [Editor Report · Decision Letter 1]

23 Oct 2023

PONE-D-23-21279R1Suicidal ideation, attempt, and its associated factors among Adult HIV/AIDS patients in Ethiopia: a systematic review and meta-analysis studyPLOS ONE

Dear Dr. Bogale,

Thank you for submitting your manuscript to PLOS ONE. After careful consideration, we feel that it has merit but does not fully meet PLOS ONE’s publication criteria as it currently stands. Therefore, we invite you to submit a revised version of the manuscript that addresses the points raised during the review process.

We look forward to receiving your revised manuscript.

Kind regards,

Chalachew Kassaw Demoze, Msc

Academic Editor

PLOS ONE

Journal Requirements:

**Additional Editor Comments:**

Dear author , You revised the manuscript as per the reviewer's comment.

I do have a concern-associated factor, which is not statistically significant because it includes 1. Therefore, I remind you to look carefully, analyze again, and write a statistically significant variable.

1.62 0.85, 2.39

1.97 0.81, 3.13

1.31 0.77, 1.84

1.69 1.03, 2.35

1.78 1.07, 2.50

0.98 0.13, 1.84

1.08 0.24, 1.91

---

## [Author Response · Author response to Decision Letter 1]

24 Oct 2023

RESPONSE TO EDITOR AND REVIEWERS

RESPONSE TO EDITOR

Response: Thank you for coordinating the review process and making fruitful comments. We have revised the manuscript and addressed the reviewer's comments.

Journal Requirements: 

Comment: Journal Requirements:

1. Please ensure that your manuscript meets PLOS ONE's style requirements, including those for file naming. The PLOS-ONE style templates can be found at

RESPONSE: We have checked and attest that all formatting and style requirements have been met and revised based on the guideline 

Comment: 2. Please review your reference list to ensure that it is complete and correct. If you have cited papers that have been retracted, please include the rationale for doing so in the manuscript text, or remove these references and replace them with relevant current references. Any changes to the reference list should be mentioned in the rebuttal letter that accompanies your revised manuscript. If you need to cite a retracted article, indicate the article’s retracted status in the References list and also include a citation and full reference for the retraction notice.

Response: Thank you for your fruitful comments. We have reviewed our reference list, and we found it complete and correct.

COMMENT: Additional Editor Comments:Dear Author , It is good work

Here are some of the concerns about a paper:

1. First, it is better to re-write a manuscript with correct alpahabetical and grammatical revisions

Response: Thank you for your kind words and fruitful comments. We have made extensive revisions to our manuscript to reduce alpahabetical and grammatical errors.

 COMMENT: 2. it is better to state a review article on the introduction part of the manuscript , I mean, you only stated a primary study

Response: Thank you for your kind words and fruitful comments. We have revised it by adding the results of the study from a systematic review and meta-analysis.

COMMENT: 3. It is better not to merege sub group analysis (Amhara vs Harar); rather, it is better to do regional vs administrative city analyses.

RESPONSE: Thank you for your kind words and fruitful comments. We have seen it, and when we see our data, we have 3 studies from the Amhara Region and only 1 study from Harer. If we classify our subgroup analysis into 3 groups, such as administrative city analysis, Amhara region, and Harer region, we cannot address the pooled prevalence in Harer region since it is a single study. So that, we try to categorize studies into two based on their homogeneous characteristics, and we assume that all cities from each region in Ethiopia have comparable urbanization and infrastructure. So that it is better to categorize it into two categories: administrative city and city from regions. I kindly ask that you consider this during the review.

Comment: 4. On the discussion part of the manuscript, please discuss the review articles done between 2013 and 2023.

Response: Thank you for your kind words and fruitful comments. We have revised it based on your comment.

Additional Editor Comments:

COMMENT: Dear author , You revised the manuscript as per the reviewer's comment.

I do have a concern-associated factor, which is not statistically significant because it includes 1. Therefore, I remind you to look carefully, analyze again, and write down a statistically significant variable.

1.62 0.85, 2.39

1.97 0.81, 3.13

1.31 0.77, 1.84

1.69 1.03, 2.35

1.78 1.07, 2.50

0.98 0.13, 1.84

1.08 0.24, 1.91

Response: Thank you for your kind words and fruitful, constructive follow-up comments. We have revised it based on your comment, and we rewrite it again in detail.

Response to Reviewer 1:

Dear Authors,

Thank you for your contribution in the field. I have had the opportunity to review your manuscript and found it well written. However, there are some points you need to consider, and I have mentioned them section by section as follows:

RESPONSE: We thank the reviewer for kind words. We have revised our manuscript based on the comments as described below.

Abstract:

REVIEWER COMMENT:  Background: The background section of the abstract should focus on stating the problem rather than defining suicidal attempts and ideation, which can be done under the main text. It is good to clearly state the aim or objective of the review.

Response: We thank the reviewer for fruitful comments. We have revised our manuscript based on the comments by deleting the definition of suicidal attempt and ideation, focusing on stating the problem, and adding sentences that describe the aim or objective of the review. 

REVIEWER COMMENT: Results: The result section should highlight the main findings in line with the objective of the study. Accordingly, factors associated with suicidal ideation and/or attempts should be discussed. In addition, a detailed description of the area-specific pooled prevalence might not be very important in the abstract. Rather, you can describe if there is significant geographical heterogeneity.

Response: Thank you for your fruitful comments. We have revised it by deleting unnecessary detail and highlighting the main findings in line with the objective of the study, including factors associated with suicidal ideation and/or attempt.

REVIEWER COMMENT: Conclusion: “The pooled magnitude of suicidal ideation and attempt was high, and factors like advanced stages of AIDS, co-morbid depression, perceived stigma, and poor social support were related to it.” From this statement, it is not known whether the factors are associated with suicidal ideation or attempts. Thus, please make it specific. 

Response: Thank you for your fruitful comments. We have revised it to make it clear.

Introduction:

REVIEWER COMMENT: The introduction is well framed. Yet, there are some points you need to consider. 1. Your population is PLWIHV. Thus, a detailed statement of the problem regarding the general population would not have much significance. Accordingly, please highlight the burden in general briefly and focus on the burden among HIV people.

RESPONSE: thank you for fruitful comments. We have revised it based on your comments.

Methods:

REVIEWER COMMENT: Data extraction: Was the data extraction done independently by the mentioned authors? What was the remedy if there is a discrepancy between the extracted data? 

RESPONSE: thank you for fruitful comments. We have take the following remedy if a discrepancy between the extracted data were occurred:- During the extraction process, discrepancy between the extracted data were resolved through logical discussion among the three authors, and the final consensus was approved with the participation of authors (NK and HBE).

REVIEWER COMMENT: Similarly, for the quality assessment how was it performed?

RESPONSE: Thank you for fruitful comments. The quality assessment was performed as follow:-

The Newcastle-Ottawa quality assessment scale [16] was used to rate the caliber of the research that was considered for the final analysis. This scale allows for possible scores ranging from 0 to 10. On this scale, a score of 8 or more indicated good quality, a score of 3 to 7 indicated moderate quality, and a score of less than this indicated low quality. The quality of the included studies were evaluated by two independent authors (EKB and TF) using the Newcastle-Ottawa quality assessment scale [16]. The third author (NK) discussed and resolved any disagreements. Out of the seven studies that were considered, two had moderate quality and five had good quality.

Results:

REVIEWER COMMENT: Identification of studies: On the PRISMA flowchart, please identify the number of articles identified from each data base. In addition, include the reason of exclusion for the 287 articles in the chart.

RESPONSE: Thank you for fruitful comments. We have revised it based on your comments. We have included the number of articles identified from each data base and we have provide reasons of exclusion for the 287 articles in the chart.

REVIEWER COMMENT: It is good to do the subgroup analysis. However, how did you treat the studies conducted in Amhara and Harar together? This is because the heterogeneity with regard to the suicidal ideation still exists when the studies are combined together or how do you explain the persistent heterogeneity?

RESPONSE: Thank you for your kind word and fruitful comments. We have seen it and when we see our data we have 3 studies from Amhara Region and only 1 study from Harer. If we classify our subgroup analysis in to 3 such as adminstarative city analaysis, Amhara region and Harer region, we can not address the pooled prevalence in Harer Region since it has a single study. So That, we try to categorize studies in to two based on their homogeneous characteristics and we assumed that all cities from each region in Ethiopia have comparable urbanization and infrascture. So that it is better to categorize it in two as adminstarative city and city from Regions. We have checked that the source of heterogeneity with regard to the suicidal ideation still exists due to the studies are combined together or not, by re analysing our data by excluding Harer region from the group and we got I2 (%)= 97.33. So that, please try to noticed that persistent heterogeneity in this study not come from combining two regions in one category. It might be due to unexplained other reasons. I kindly ask you to consider this during review.

REVIEWER COMMENT: - replace P=0.0000 by P<0.001 Please, replace P=0.0000 by P<0.001.

RESPONSE: Thank you for fruitful comments. We have revised it by replacing P=0.0000 by P<0.001

REVIEWER COMMENT:- Please, avoid repetitions; for example, the following statements were repeated a number of times….. “The seven studies included in the meta-analysis, three were from Addis Ababa administrative councils [11, 12, 18], and four were from cities found in two regions, Amhara and Harar [10, 13, 19, 20].”

RESPONSE: Thank you for fruitful comments. We have revised it by deleting redundant paragraph as mentioned above.

REVIEWER COMMENT: For the factors associated with the suicidal ideation and attempt, it is good to just describe the pooled effect sizes than the study specific effect sizes.

RESPONSE: Thank you for fruitful comments. We have revised our manuscript by describing the pooled effect sizes rather than the study specific effect sizes. 

Discussion:

REVIEWER COMMENT: The first paragraph of a discussion should be the highlight of the current study and the main findings. Thus please, describe the highlight of your study and the main findings and then discuss each finding one by one. Please, make sure that you compare your finding with comparable studies i.e., systematic reviews and meta-analyses.

RESPONSE: Thank you for fruitful comments. We have revised it based on your comment by focusing highlighting of the current study and the main findings as first paragraph of a discussion as following paragraph and then we discussed each finding one by one. Moreover, we make sure for you that we compared our finding with other comparable studies such as systematic reviews and meta-analyses.

In this systematic review and meta-analysis study, seven studies that were conducted between 2013 and 2023 with 2593 participants were analyzed to estimate suicidal ideation, suicidal attempts, and its associated factors among adult HIV/AIDS patients in Ethiopia. All seven studies assessed both suicidal ideation and suicidal attempts as a primary outcome variable. According to this metaanalysis, the pooled prevalence of suicidal ideation and suicidal attempt was 22.1%, and 11.1, respectively. Living alone, having comorbidity or other opportunistic infection, having WHO clinical stage III of HIV, having WHO clinical stage IV of HIV, having co-morbid depression, having perceived HIV stigma and having family history of suicidal attempt were significantly associated with suicidal Ideation. Being female, having opportunistic infections, WHO clinical stage III of HIV, co-morbid depression, having poor social support, and having WHO clinical stage IV were significantly associated with suicidal attempt.

REVIEWER COMMENT:….. “Our subgroup analysis showed that the pooled prevalence of suicidal ideation among adult HIV/AIDS patients was 24.2% in Addis Ababa administrative councils, higher than the pooled prevalence of suicidal ideation in the city from the regions (Amhara and Harar) of 20.7%. This difference might be due to variations in the level of urbanization.” Please, explain how urbanization is related to suicidal ideation.

RESPONSE: Thank you for fruitful comments. In our manuscript, we provide explanation for how urbanization is related to suicidal ideation as follow:-

When we compare cities found in the region and Addis Ababa administrative councils, the level of urbanization is high in Addis Ababa. Numerous issues may arise as a result of urbanisation, including high population density, concentrated poverty, alienation from nature, increased noise and air pollution, and social isolation [31]. Common mental disorders are being diagnosed more frequently, and self-harm and suicide are becoming more common, all of which are related to the general trend towards urban living (32).

Conclusion:

REVIEWER COMMENT: Conclusion is about the implication of your study than the mere figure of the results. Accordingly, please, describe the implication of the pooled prevalence of suicidal ideation and attempt than the proportion itself. 

RESPONSE: We thank the reviewer for kind words and fruitful comments. We have revised it by describing the implication of the pooled prevalence of suicidal ideation and attempt than the proportion itself.

---

## [Editor Report · Decision Letter 2]

25 Oct 2023

Suicidal ideation, attempt, and its associated factors among Adult HIV/AIDS patients in Ethiopia: a systematic review and meta-analysis study

PONE-D-23-21279R2

Dear Dr Eyob 

We’re pleased to inform you that your manuscript has been judged scientifically suitable for publication and will be formally accepted for publication once it meets all outstanding technical requirements.

Kind regards,

Chalachew Kassaw Demoze, Msc

Academic Editor

PLOS ONE

Additional Editor Comments (optional):

Dear author , it is good that you revised accordingly

I want to notice some issues

1. Please carefully revise the spelling and grammar. Still, there are a lot of these problems.

2. Please revise a discussion and describe it precisely.
---

## [Editor Report · Acceptance letter]

2 Mar 2024

PONE-D-23-21279R2 

PLOS ONE

Dear Dr. Bogale, 

I'm pleased to inform you that your manuscript has been deemed suitable for publication in PLOS ONE. Congratulations! Your manuscript is now being handed over to our production team.

Kind regards, 

on behalf of

Mr Chalachew Kassaw Demoze 

Academic Editor

PLOS ONE